# Multitarget Potential of Phytochemicals from Traditional Medicinal Tree, *Terminalia arjuna* (Roxb. ex DC.) Wight & Arnot as Potential Medicaments for Cardiovascular Disease: An In-Silico Approach

**DOI:** 10.3390/molecules28031046

**Published:** 2023-01-20

**Authors:** Vikas Kumar, Nitin Sharma, Raha Orfali, Chirag N. Patel, Radwan Alnajjar, Rakshandha Saini, Anuradha Sourirajan, Prem Kumar Khosla, Kamal Dev, Shagufta Perveen

**Affiliations:** 1University Institute of Biotechnology, Chandigarh University, Mohali 140413, Punjab, India; 2Faculty of Applied Sciences and Biotechnology, Shoolini University of Biotechnology and Management Sciences, Bajhol, Solan 173229, India; 3Department of Biotechnology, Chandigarh Group of Colleges, Mohali 140307, Punjab, India; 4Department of Pharmacognosy, College of Pharmacy, King Saud University, P.O. Box 2457, Riyadh 11451, Saudi Arabia; 5Department of Botany, Bioinformatics, Climate Change Impacts Management, School of Sciences, Gujarat University, Ahmedabad 380009, Gujarat, India; 6Chemical Biology Laboratory, Center for Cancer Research, National Cancer Institute, National Institute of Health, Frederick, MD 21702, USA; 7Department of Chemistry, Faculty of Science, University of Benghazi, Benghazi 16063, Libya; 8Department of Pharmacology and Toxicology, Wright State University, Dayton, OH 4543, USA; 9Department of Chemistry, School of Computer, Mathematical and Natural Sciences, Morgan State University, Baltimore, MD 21251, USA

**Keywords:** *Terminalia arjuna*, cardiovascular diseases, phytochemicals, molecular docking, MD simulations, MM/GBSA

## Abstract

Cardiovascular diseases (CVDs) are the leading cause of mortality worldwide. *Terminalia arjuna* (Roxb. ex DC.) Wight & Arnot of the Combretaceae family is one of the most frequently approved and utilized medicinal trees in the traditional medicinal system, which was used for the treatment of a variety of diseases, including cardiovascular disorders. The present study aims to identify phytochemicals from *T. arjuna*, that do not exhibit any toxicity and have significant cardioprotective activity using an in-silico technique. Four different cardiovascular proteins, namely human angiotensin receptor (PDB ID: 4YAY), P38 mitogen-activated protein kinase (MAPK, PDB ID: 4DLI), 3-hydroxy-3-methylglutaryl-coenzyme A (HMG-Co A) reductase (PDB ID: 1HW9), and human C-reactive protein (PDB ID: 1B09), were used as target proteins to identify potential inhibitors using a virtual screening of the phytochemicals in *T. arjuna* revealed casuarinin as a potential inhibitor of all selected target proteins with strong binding energy. Furthermore, MD simulations for a 100 ns time scale also revealed that most of the key protein contacts of all target proteins were retained throughout the simulation trajectories. Binding free energy calculations using the MM-GBSA approach also support a strong inhibitory effect of casuarinin on target proteins. Casuarinin’s effective binding to these proteins lays the groundwork for the development of broad-spectrum drugs as well as the understanding of the underlying mechanism against cardiovascular diseases through in vivo and clinical studies.

## 1. Introduction

Cardiovascular diseases (CVDs) are a category of heart and blood vessel disorders that include coronary heart disease, cerebrovascular disease, peripheral arterial disease, rheumatic heart disease, congenital heart disease, deep vein thrombosis, and pulmonary embolism [1]. CVDs are a leading cause of death in both developed and developing nations (https://www.who.int/news-room/fact-sheets/detail/cardiovascular-diseases, accessed on November 19, 2022). CVDs are predicted to kill 18.6 million people per year, accounting for 33% of all global deaths. More than 75% of CVD deaths occur in low- and middle-income nations [2]. The main causes of the rise in CVDs are urbanization and lifestyle changes. More than 23.3 million individuals are predicted to die each year from CVDs by 2030 [3].

The renin-angiotensin system (RAS) is a critical regulator of cardiovascular and renal function [4]. Blockage of RAS using angiotensin-converting enzyme (ACE) inhibitors and angiotensin receptor antagonists has become a first-line treatment for hypertensive target organ damage and progressive renal disease [4,5]. High blood cholesterol levels are another major contributor to the progression of coronary artery disease (CAD) [6]. Cholesterol is produced through the mevalonate pathway. In cells, the concentration of mevalonate is tightly controlled through the activity of 3-hydroxy-3-methylglutaryl-CoA reductase (HMG-CoA reductase) [7]. HMG-CoA reductase is an enzyme that plays a role in the biosynthesis of endogenous cholesterol in the liver. It catalyzes the conversion of HMG-CoA to mevalonate, a precursor of sterols including cholesterol. Inhibition of HMG-CoA reductase decreases mevalonate production, therefore reducing cholesterol synthesis [8]. Several large-scale clinical trial studies showed that inhibition of this enzyme could significantly reduce cholesterol levels and reduce the risk of stroke and mortality by 29% and 22%, respectively [9,10].

P38 kinases, which are members of the mitogen-activated protein kinase (MAPK) family, play an important part in the signal transduction cascade causing post-ischemic cardiac apoptosis. RAS angiotensin II suppressed adipogenesis by activating the ERK 1/2 and MAPK kinase pathways [11] and MAPK p38 inhibition may reduce reperfusion injury [12]. Chronic insulin exposure and metabolic stressors in ischemic conditions activate p38 MAPK and enhance insulin receptor substrate 1/2 (IRS1/2) degradation, culminating in AKT inactivation and eventual myocyte death and heart failure [13]. C-reactive protein (CRP), an acute-phase protein, is not only an inflammatory marker but also a direct cause of CVD; hence, interventions that lower CRP should be beneficial for both primary and secondary CVD prevention [14,15]. A number of side effects were observed in synthetic medications used to treat various cardiovascular problems. Hyperkalemia, dry cough, angioedema, hypotension, dizziness, headache, and renal failure are the most common side effects of ACE inhibitors (ACEi) [16,17]. Another serious side effect of ACEi is angioedema. It can affect any region of the body, including the intestine, but edoema of the tongue, glottis, and/or larynx is the most serious, causing airway obstruction [18]. The most prevalent side effect of statin therapy for the treatment of cardiovascular disorders is myalgia (1–10%) [19].

Plant-based medical systems have grown in popularity in the modern era due to their low cost, ease of availability, effectiveness, and lack of adverse effects [20]. *Terminalia arjuna* (Roxb) Wight and Arnot (*T. arjuna*), also called as “Arjuna”, has been used as a cardiotonic in heart failure, ischemic heart disease, cardiomyopathy, atherosclerosis, and myocardial necrosis [21,22,23]. Several studies have been conducted to investigate the cardioprotective potential of *T. arjuna* stem bark using various models [24,25,26,27,28]; however, the mechanism of action of *T. arjuna* phytochemicals against cardiovascular target proteins remains unknown. Mythili et al. [29] discovered that arjunolic acid, found in *T. arjuna* stem bark, protects against CsA-induced cardiotoxicity.

As increasingly improved technologies for looking for pharmaceuticals derived from phytochemicals present in a range of medicinal plants have been available, computer-aided drug discovery approaches have emerged [30,31]. Computational prediction models are critical in influencing strategy choices in technological and pharmaceutical research. They have also been used to predict in silico pharmacokinetic, pharmacological, and toxicological performance [32,33]. Molecular docking tools, MD simulations, and ADMET prediction were used in a new strategy that was developed to find the interaction between the phytochemicals from *T. arjuna* and synthetic drugs (phosphocholine, simvastatin, lisinopril, losartan, and losmapimod) and cardiovascular targets. This strategy was developed as part of an effort to expand the scope of our investigation into cardiovascular disorders.

## 2. Results

### 2.1. Molecular Docking Analysis

Table 1 summarizes the binding energies of selected phytocompounds with target proteins using the Glide (grid-based ligand docking) program. Among all phytocompounds, casuarinin had a good binding affinity and better binding modes than selected phytocompounds and standard drugs. Casuarinin showed binding energies of −9.678, −10.685, −9.216, and −18.276 kcal mol^−1^ with 1B09, 1HW9, 4DLI, and 4YAY proteins, respectively (Table 1). The redocking was also performed to check the binding interaction of native ligands with the respective targets and it was observed that the docked phytochemicals showed better binding than that of re-docked ligands (Appendix A).

Casuarinin binding interactions were examined using the Discovery Studio (DS) visualizer (Figure 1A–H) and were found to have six hydrogen bonds with Tyr(A):35, Arg(A): 167, Phe(A): 182, Tyr(A): 184, Asp(A): 263, Gln(A): 267 residues of 4YAY protein (Figure 1A&B), three hydrogen bonds with Lys(A): 249, Ser(A): 251, Arg(A): 256 residues of 4DLI protein (Figure 1C&D), three hydrogen bonds with Asn(A): 567, Arg(A): 571, Glu(A): 719 residues of 1HW9 protein (Figure 1E&F) and two hydrogen bonds with Ala(A): 92, and Asp(A): 112 residues of 1B09 protein (Figure 1G&H).

### 2.2. MD Simulations Study

Casuarinin was found to have the best binding energy with all selected target proteins; therefore, complexes of casuarinin with 4YAY, 4DLI, 1HW9, and 1B09 proteins were further selected for MD simulations for 100 ns to study their protein–ligand interactions. When performing MD simulations, the root mean square deviation (RMSD) is used to measure the average change in displacement of a selection of atoms for a particular frame with respect to a reference frame. It is calculated for all frames in the trajectory. The plots in Figure 2 show the RMSD evolution of a protein (left *y*-axis). The docked pose of ligand and protein as a whole complex is considered the reference starting frame, and then the movement from this reference position during MD simulation is measured by aligning all the protein frames obtained during the MD trajectories. For the complexes of casuarinin with 4YAY (Figure 2A), casuarinin-4DLI (Figure 2B), and casuarinin-1HW9 (Figure 2C), the protein backbone hovers around the value of RMSD not exceeding 4.8 Å, and for the casuarinin-1B09 complex (Figure 2D), the value of RMSD stays well under 2.8 Å. Ligand RMSD (right *y*-axis, plots in Figure 2) indicates the stability of ligand posture in relation to the docked position of the ligand in the protein’s binding cleft. “Lig fit Prot” suggests the RMSD of a ligand for protein backbone. For this, the values slightly larger than the protein’s RMSD are considered satisfactory, but if the values observed are significantly larger than the protein’s RMSD, then it is likely that the ligand acquires a different stable position than the original posture. For casuarinin-4YAY complex (Figure 2A), the Lig fit Prot stays significantly lower than protein’s RMSD from 0–42 ns and after 70 ns throughout the simulation, suggesting slight changes in pose between 42 and 70 ns thereafter, the orientation of ligand remains stable. For the casuarinin-4DLI complex (Figure 2B), the Lig fit Prot stays significantly lower than the protein’s RMSD throughout the simulation, suggesting that the orientation of the ligand remains the same. For the casuarinin-1HW9 complex (Figure 2C), the Lig fit Prot value stabilizes after 60 ns, suggesting the casuarinin changes poses up to 60 ns and then stabilizes to a constant pose. For the casuarinin-1B09 complex (Figure 2D), the Lig fit Prot value stabilizes between 20 and 60 ns, and after 80 ns throughout the stimulation, suggesting the casuarinin remains stable up to 60 ns and then slightly changes pose and stabilizing to a constant pose after 80 ns.

The root mean square fluctuation (RMSF) is useful for portraying confined changes along the protein chain (Figure 3). In the graph, the peaks demonstrate regions of the protein that vary the most throughout the simulation. Ordinarily, the tails (N-and C-terminal) change more than other internal regions of the protein. Secondary protein regions such as alpha helices and beta strands are generally more inflexible and rigid than unstructured regions and thus vacillate, not exactly like loop-forming protein regions. α-helical and β-strand areas are featured in red and blue foundations separately. These districts are characterized by helices or strands that endure over 70% of the whole re-enactment. Protein deposits that contact ligand is set apart with green-hued vertical bars. The RMSF of the protein can likewise be related to the exploratory x-beam B-factor (right Y-hub). Because of the distinction between the RMSF and B-factor definitions, balanced correspondence ought not to be normal. Notwithstanding, the reproduction results should resemble crystallographic information. It is seen that both buildings of casuarinin with 4YAY (Figure 3A), 4DLI (Figure 3B), 1HW9 (Figure 3C), and 1B09 (Figure 3D) and trends of RMSF and B-factor definitions correspond similarly in all protein–ligand complexes.

The interaction of casuarinin with target proteins during the whole course of simulation appears in Figure 4A–D. The RMSD of a ligand with respect to the reference compliance (ideally, the first frame of the trajectory is used as the reference, and it is viewed as time *t* = 0). The radius of gyration (rGyr) evaluates the “extendedness” of a ligand and is equal to its essential snapshot of idleness. Intramolecular hydrogen bonds (intraHB) show the number of inner hydrogen bonds inside a ligand atom. Molecular surface area (MolSA) portrays the sub-atomic surface figure with a 1.4 Å test sweep. This value is proportionate to a van der Waals surface zone. Solvent accessible surface area (SASA) is the surface zone of a molecule open for access to water molecules. Every one of these highlights is the quality of an individual compound (ligand); subsequently, these estimations of two unique ligands cannot be compared directly.

Figure 5 represents how the ligand behaved while interacting with the target proteins during MD simulation for 100 ns. Protein interactions with the ligand can be monitored throughout the simulation. These interactions can be categorized by type and summarized, in Figure 5A for the casuarinin-4YAY complex, Figure 5B for the casuarinin-4DLI complex, Figure 5E for casuarinin-1HW9 complex, and Figure 5F for the casuarinin-1B09 complex. There are four types of protein–ligand interactions: hydrogen bonds, hydrophobic interactions, ionic interactions, and water bridges. Every connection type contains more explicit subtypes, which can be investigated through the “simulation interactions diagram” board. The stacked bar outlines are consistent throughout the direction; for example, an estimate of 0.8 suggests that collaboration be maintained for 80% of the simulation time. Qualities greater than 1.0 are possible because some protein aggregates may make multiple contacts of the same subtype with the ligand. A timetable portrayal of the associations and contacts (hydrogen bonds, hydrophobic, ionic, water spans) is shown in Figure 5C for the casuarinin-4YAYcomplex, Figure 5D for the casuarinin-4DLI complex, Figure 5 for the casuarinin-1HW9 complex, and Figure 5H for the casuarinin-1B09 complex. These figures depict which deposits communicate with the ligand in every direction. A few residues make more than one explicit contact with the ligand, which is shown by a hazier shade of orange, as indicated by the scale on one side of the plot. These plots are very crucial, suggesting the interaction of casuarinin with amino acids of target proteins throughout the simulation and that these ligands are not dissociating away from their interacting site; however, slight variations in the RMSD and RMSF values of the ligand, as showed in Figure 2 and Figure 3 respectively, suggest that these ligands may be reorienting themselves during the simulation.

### 2.3. MM/GBSA Binding Free Energy Calculations

Post simulation analysis of all four protein–ligand complexes was performed by taking snapshots of the trajectory profiles developed on performing 100 ns MD simulations, as depicted in Table 2. Casuarinin was found to have negative ∆G binding with all target proteins. Van der Waals interactions (∆*G*_vdW_) of casuarinin with selected target proteins were found to be between −12.45 and −68.58 kcal/mol, suggesting that casuarinin tends to stay in the vicinity of the interacting amino amides of target proteins. Coulomb energy was found to be negative for all complexes, indicating that casuarinin has a low potential energy with all target proteins and suggesting that protein–ligand complexes are more stable. In addition to the total energy, the contributions to the total energy from different components such as hydrogen-bonding correction, lipophilic energy, and van der Waals energy are provided in Table 2.

### 2.4. Assessment of Drug Likeness and Toxicity Prediction

Lipinski’s rule of five and the ADMET prediction of the top-ranked compound were studied to understand the amenability of pharmacokinetic properties and toxicity properties. Lipinski’s rule of five and the toxicity prediction of casuarinin and drugs are shown in Table 3. Casuarinin was found to have drug-likeness violations, but toxicity parameters were successfully met with a lack of hepatotoxicity, carcinogenicity, and cytotoxicity as that of drugs. The predicted LD_50_ (mg/kg) for casuarinin was 2170; hence, it was categorized as toxicity class-5 by Protox-II. However, the predicted LD_50_ (mg/kg) for drugs with toxicity classes III-VI was found to be between 300 and 12,900. 

## 3. Discussion

In pharmaceutical research, computational strategies are of great value as they help in the identification and development of novel promising compounds especially by molecular docking methods [34,35]. These methods have been used by various research groups to screen potential novel compounds against a variety of diseases [36]. Angiotensin-converting enzyme (ACE) has a significant role in the regulation of blood pressure and ACE inhibition with inhibitory peptides is considered a major target to prevent hypertension [37]. Several studies have used a docking approach to inhibit the expression of the ACE protein with natural compounds. Quercetin glycosides showed optimum binding affinity with angiotensin-converting-enzyme (−8.5 kcal mol^−1^) as compared to enalapril (−7.0 kcal mol^−1^), thereby indicating the role of quercetin glycosides as a potential candidate to treat hypertension, myocardial infarction, and congestive heart failure [38]. Similarly, methyl gallate and quercetin 3-*O*-β-D-glucopyranosyl-(1‴−6″)-α-rhamnoside from *Phyllanthus niruri* herb have been reported as potent ACE inhibitors [39]. Impertonin from whole fruit extract of *Aegle marmelos* showed strong interaction with HMG-CoA reductase enzyme [40]. Secondary metabolites such as dichloroacetic acid 2, 2-dimethylpropyl ester, 1, 6, 10-dodecatriene-3-ol, 3, 7, 11-trimethyl-[S-(Z)]-, isopropyl acrylate, and 3, 3-dimethylacryloyl chloride formed strong binding with active sites of HMG CoA reductase [41].

Talapatra et al. [42] reported that phytocompounds from *Calotropis procera* such as methyl myrisate (−3.0 kcal mol^−1^) and methyl behenate (−3.2 kcal mol^−1^), β-sitosterol (−5.6 kcal mol^−1^), uzarigenin (−5.5 kcal mol^−1^) and anthocyanins (−5.4 kcal mol^−1^) showed good binding with CRP receptors. Caffeic acid had remarkable interaction with proteins involved in inflammatory response (COX-2, COX-1, FXa and integrin αIIbβIII), thereby, having the potential to be developed as cardiovascular-safe anti-inflammatory medicine [43]. Khan [44] studied the interactions between 4YAY (Angiotensin-I) receptor protein with phytocompounds from *Alangium salvifolium.* The compound alangum 1 [Alangium1(4(benzoyloxy) methyl-2hydroxyphenoxy tetrahydorxy hexoxone 1,2,3,4,5, pentaium] showed the best glide docking XP score −8.5 kcal/mol binding energy value with best fit simulation study. Study conducted by Liu et al. [45] identified novel ACE inhibitory peptides Ala-Val-Lys-Val-Leu (AVKVL), Tyr-Leu-Val-Arg (YLVR), and Thr-Leu-Val-Gly-Arg (TLVGR) with IC_50_ values of 73.06, 15.42, and 249.3 μM, respectively. All peptides inhibited the ACE activity via a non-competitive mode. The binding free energies of AVKVL, YLVR, and TLVGR for ACE were −3.46, −6.48, and −7.37 kcal mol^−1^, respectively. Docking of HMG-CoA (PDB ID: 3CCZ) with guajavarin, was found to be least binding energy (−100.092 kcal mol^−1^) resulted in formation of four hydrogen bonds with the residues and amino acids SER 684 (3.2 Å), LYS692 (2.4 Å), ASP 690 (2.7 Å), and LYS 691 (3.1 Å) respectively [46]. Study on phytocompounds from *T. arjuna* with phosphodiesterase 5A, sodium-potassium pump, and β-adrenergic receptor showed that casuarinin showed multiple inhibitions on phosphodiesterase 5A and sodium-potassium pump, whereas pelargonidin on phosphodiesterase 5A and β-adrenergic protein targets [47]. Recently, Murad et al. [48] reported binding of curcumin, quercetin, resveratrol, and eucalyptol with active sites of chemokine (C-X-C motif) receptor-4 (CXCR4) and CXCR7 receptors. Although all compounds demonstrated drug-like properties, but eucalyptol has promising potential because it can be used by inhalation or skin massage.

## 4. Materials and Methods

### 4.1. Retrieval of Proteins

With the help of a literature review, we narrowed our focus to four proteins that have been linked to different cardiovascular disorders (Table 4). In order to better understand how our targets work, we accessed their three-dimensional crystal structures from the RCSB protein data library (www.rcsb.org, accessed on 10 November 2022) (Figure 6). The retrieved structures were pre-processed using Protein Preparation Wizard of Schrodinger suite (Academic licence, Schrodinger Suite, 2014 founded by Richard A. Friesner, and William A. Goddard III, New York, NY, USA). Furthermore, restrained minimization was carried out to obtain a geometrically stable protein conformation. The receptor grid was generated at the ligand-binding site of selected protein targets by selecting the position of the co-crystal ligands. This defined grid in the receptor structure was used as the docking site for the virtual screening of selected phytocompounds. The details of grid box coordinates of selected target proteins are summarized in Table 4.

### 4.2. Ligand Preparation

The selected phytochemicals and synthetic drugs were obtained from Pubchem (https://pubchem.ncbi.nlm.nih.gov/, accessed on 10 November 2022). These selected molecules were pre-processed and conformers were generated using Schrodinger Ligprep (LigPrep version 3.2, Schrodinger in 2014 by Richard A. Friesner, and William A. Goddard III, New York, NY, USA). Pre-processing of ligands included the following tasks: conversion of 2D structures to 3D format, addition of hydrogen atoms, generation of tautomers and ionization states, neutralization of charged groups, and finally geometry optimization of the molecule using OPLS 2005 force field [49] (Table 5).

### 4.3. Molecular Docking

Molecular docking of selected phytochemicals of *T. arjuna* with target proteins was performed using Glide (grid-based ligand docking) program incorporated in the Schrödinger molecular modeling package with extra precision (XP). Extra-precision (XP) docking and scoring is a more powerful and discriminating method that takes longer to perform than SP. XP is intended to be utilized on ligand postures that have been proven to be high-scoring utilizing SP docking. XP also implements a more complex scoring algorithm that is “harder” than the SP GlideScore, with higher criteria for ligand–receptor form complementarity. This filters out erroneous positives that SP allows through. Because XP can penalize ligands that do not match well to the particular receptor conformation employed, we propose docking to many receptor conformations whenever feasible. The best pose based on binding energies for each ligand–protein interaction was further analyzed in Discovery Studio (DS) visualizer (Accelrys, San Diego, USA). From the interaction profile, the ligands showing high binding energy were further considered for the molecular dynamic simulations.

### 4.4. Molecular Dynamics Simulations

In order to investigate the structural stability of the receptor–ligand complexes that were formed by molecular docking, the Desmond programme version 2.0 (academic version, D. E. Shaw Research, New York, US) was utilized [50,51,52,53]. TIP3P water model with cubic periodic box including simple point charge (SPC) (10Å×10Å×10Å) was used to prepare the system [54], along with optimized potentials for liquid simulations (OPLS) all-atom force field 2005. The system was then neutralized by introducing the required amount of sodium ions. For the initial energy minimization phase and pre-equilibration in several restricted steps, receptor–ligand complexes were made available. The OPLS 2005 force field parameters, which included a relaxation time of 1 ps at constant temperature of 300 K and constant volume, were taken into account with the periodic boundary conditions in the NPT ensemble system in order to perform MD simulations [55,56]. The Smooth Particle Mesh Ewald (PME) approach (with a 10^−9^ tolerance limit), with a cut off distance of 9.0, was used to analyze protein structures every 1 ns. An average structure from the MD simulation that corresponded to the production phase was used to calculate the stability. Additionally, the histogram for torsional bonds, the radius of gyration (Rg), the root means square deviation (RMSD), and the root mean square fluctuation (RMSF) were used to examine structural alterations in relation to the dynamic role of the receptor–ligand complexes [57,58,59].

### 4.5. Binding Free Energy Calculations

The binding free energies of protein–ligand complexes have been calculated using MM-GBSA and molecular mechanics Poisson–Boltzmann surface area (MM-PBSA) [60,61]. As a result, the PRIME module of Maestro 11.4 and the OPLS-2005 force field were employed to calculate the binding energy of the best-docked ligand–receptor complex using the equation below:∆G_Bind_ = ∆E_MM_ + ∆G_Solv_ + ∆G_SA_(1)
where ∆E_MM_ is the difference of the minimized energies of the protein–ligand complex, while ∆G_Solv_ is the difference between the GBSA solvation energy of the protein–ligand complexes and the sum of the solvation energies for the protein and ligand. ∆G_SA_ indicates the surface area energies in the protein–ligand complexes and the difference in the surface area energies for the complexes [62].

### 4.6. Evaluation of Drug-Likeness and ADME/Toxicity Properties

Lipinski’s rule (rule of five, RO5) was considered the primary factor for screening of the molecules, and it was evaluated using the SWISS ADME web server (http://www.swissadme.ch/, accessed on 15 November 2022). Further, the toxicity of selected compounds was analyzed using the Protox-II tool to ascertain their risk of drugability [63]. Figure 7 shows the workflow scheme adapted in the present investigation.

## 5. Conclusions

In the present study, we targeted various cardiovascular proteins through an in-silico method using phytochemicals with a rationale to block their interactions. This approach can help block the disease target proteins to prevent cardiovascular diseases. This study found prominent interactions of casuarinin with multiple protein targets and can be used as a promising compound to treat different cardiovascular diseases. MD simulations were depicted as the best binding stability with correlative motions. Moreover, the distribution of hydrogen bonds and the energy contribution of all simulated complexes of casuarinin with target proteins were calculated through binding free energy. Most of the studies have targeted one or two cardiovascular targets, but in our study, we have selected multiple proteins, and found that casuarinin from *T. arjuna* which shows strong binding against all the proteins can be used to develop broad spectrum cardioprotective drug with no adverse effects. However, further in vitro and in vivo studies are needed to validate these results.

## Figures and Tables

**Figure 1 molecules-28-01046-f001:**
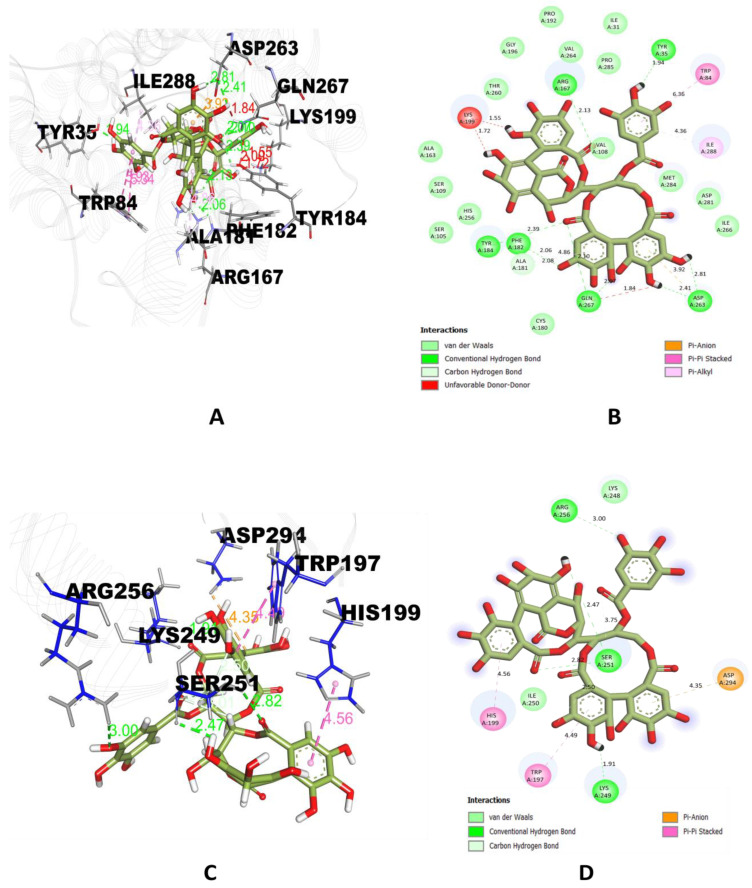
Docked pose of the top-ranked ligand (casuarinin) with target proteins. (**A**) 3-D interactions of casuarinin with interacting amino acids of 4YAY; (**B**) 2-D interactions of casuarinin with interacting amino acids of 4YAY; (**C**) 3-D interactions of casuarinin with interacting amino acids of 4DLI; (**D**) 2-D interactions of casuarinin with interacting amino acids of 4DLI; (**E**) 3-D interactions of casuarinin with interacting amino acids of 1HW9; (**F**) 2-D interactions of casuarinin with interacting amino acids of 1HW9; (**G**) 3-D interactions of casuarinin with interacting amino acids of 1B09 and (**H**) 2-D interactions of casuarinin with interacting amino acids of 1B09.

**Figure 2 molecules-28-01046-f002:**
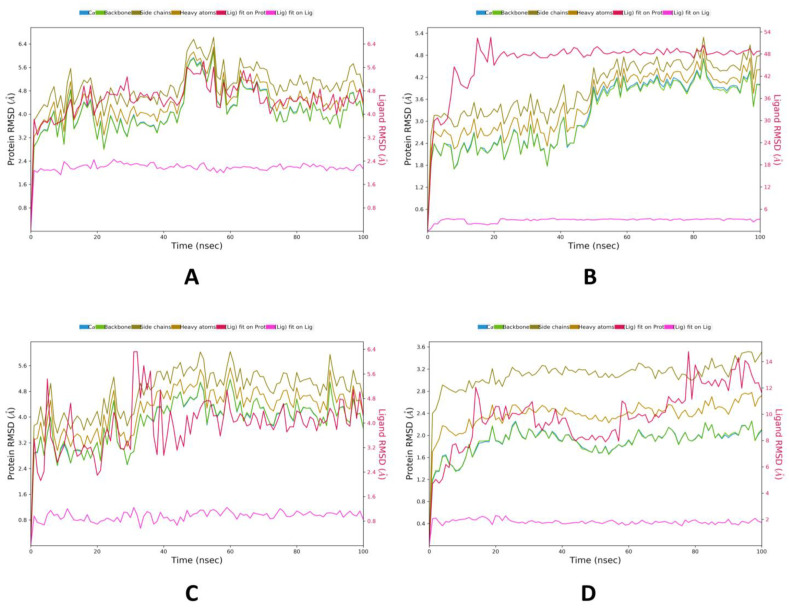
MD simulation protein–ligand interaction root-mean-square deviation (RMSD) profile of casuarinin with target proteins. (**A**) 4YAY, (**B**) 4DLI, (**C**) 1HW9, (**D**) 1B09. Color legends: Ca (blue color), side chains (green color), heavy atoms (yellow color), ligand with protein (dark pink color), ligand with ligand (pink color).

**Figure 3 molecules-28-01046-f003:**
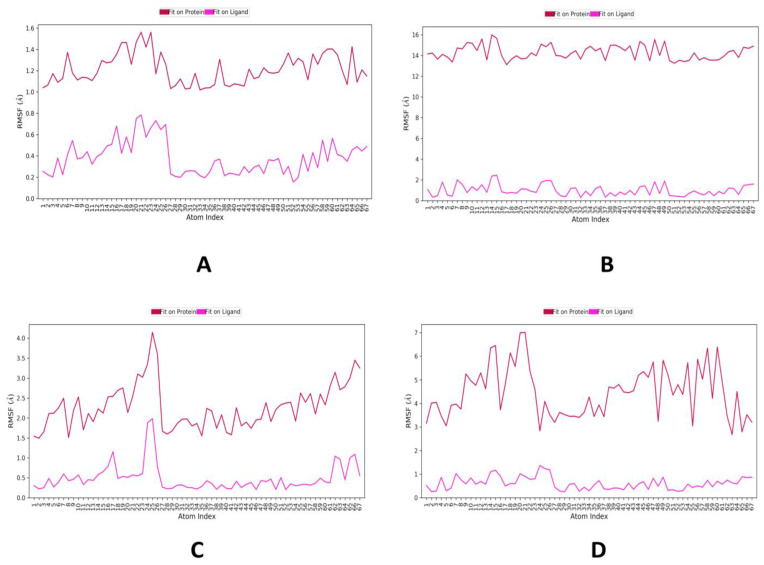
MD simulation protein–ligand interaction root-mean-square fluctuation (RMSF) profile. (**A**) 4YAY-casuarinin, (**B**) 4DLI-casuarinin complex, (**C**) 1HW9-casuarinin complex, (**D**) 1B09-casuarinin complex.

**Figure 4 molecules-28-01046-f004:**
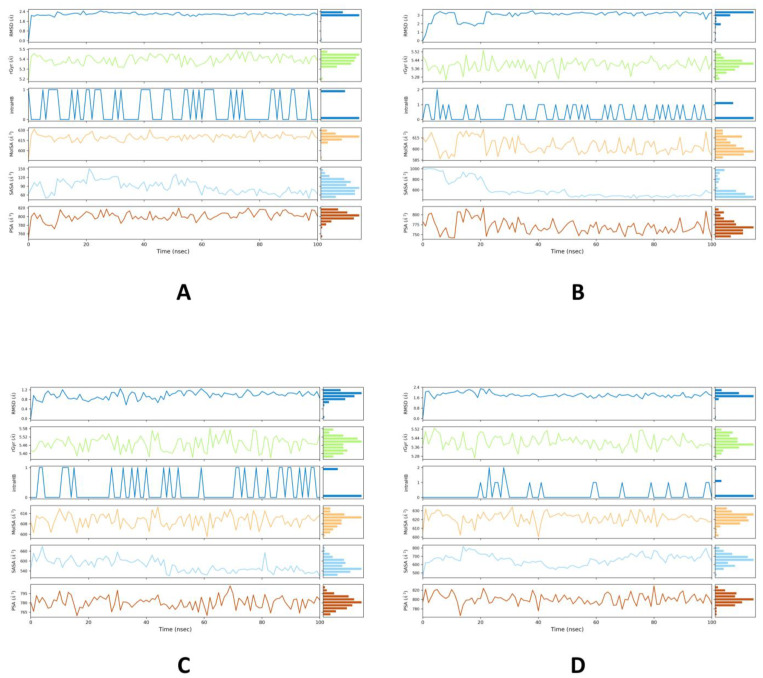
Ligand properties for casuarinin with 4YAY (**A**), 4DLI (**B**), 1HW9 (**C**), and 1B09 (**D**) proteins, such as RMSD, the radius of gyration (rGyr), intramolecular hydrogen bonds (intraHB), molecular surface area (MolSA), solvent accessible surface area (SASA), polar surface area (PSA) on interacting with protein during MD simulation.

**Figure 5 molecules-28-01046-f005:**
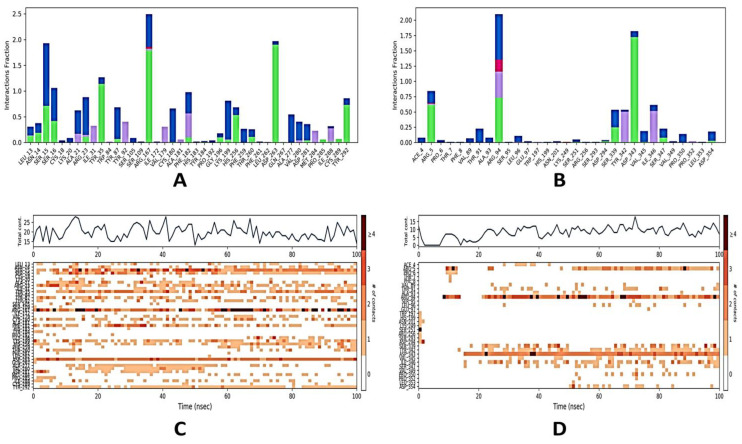
Protein–ligand interaction profile of protein–ligand complexes. (**A**) Interaction profile of crucial interacting amino acids of 4YAY with casuarinin. (**B**) Interaction profile of crucial interacting amino acids of 4DLI with casuarinin. (**E**) Interaction profile of crucial interacting amino acids of 1HW9 with casuarinin. (**F**) Interaction profile of crucial interacting amino acids of 1B09 with casuarinin. (**C**) Timeline representation of the interactions of amino acids of 4YAY with casuarinin, (**D**) timeline representation of the interactions of amino acids of 4DLI with casuarinin, (**G**) timeline representation of the interactions of amino acids of 1HW9 with casuarinin, (**H**) timeline representation of the interactions of amino acids of 1B09 with casuarinin. Different types of bar color indicate different types of bonds: hydrogen bond (green), hydrophobic contacts (purple), and water-bridge (blue). # in the graphs indicates total number of specific contacts.

**Figure 6 molecules-28-01046-f006:**
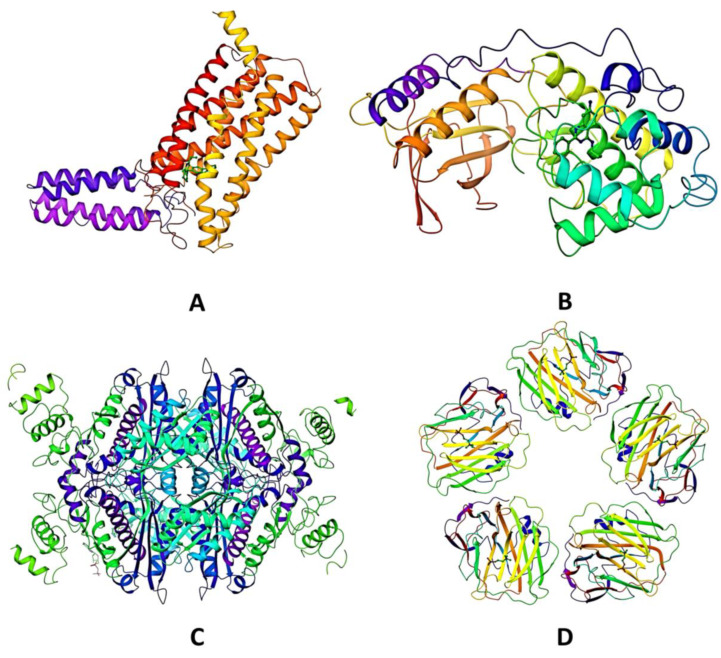
Three-dimensional structure of target proteins: (**A**) 4YAY, (**B**) 4DLI, (**C**) 1HW9, and (**D**) 1B09. Different colors of the chains indicate different types of chain-α-chains, β-pleated sheets etc. The N termini of proteins are blue in color.

**Figure 7 molecules-28-01046-f007:**
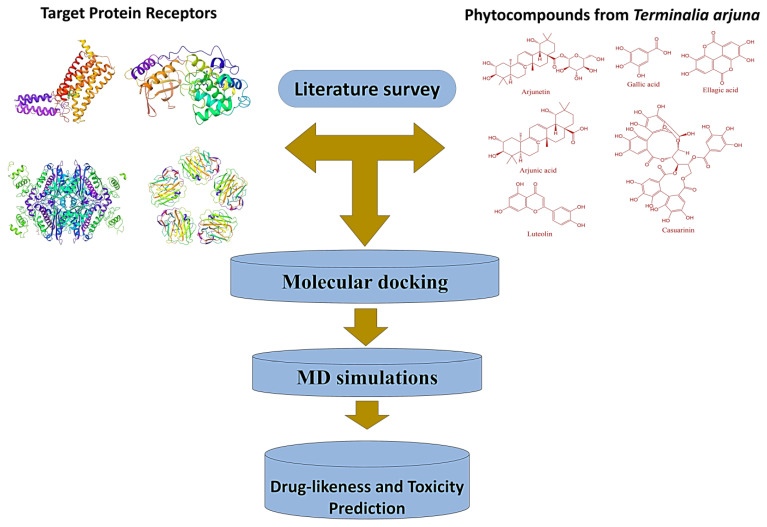
Workflow scheme adapted in the present study.

**Table 1 molecules-28-01046-t001:** Binding energy of docked phytocompounds from *T. arjuna* against targeted protein receptors. Binding energy was expressed in terms of kcal mol^−1^.

Phytocompounds	4YAY	4DLI	1HW9	1B09
3-O-Methylellagic-acid-3-rhamnoside	−7.668	−4.082	−4.411	−2.601
Arjunetin	−6.14	−3.384	−0.886	−5.307
Arjungenin	-	-	−2.534	−1.71
Arjunic acid	-	−2.774	−2.587	−1.315
Arjunolic acid	-	-	−1.812	−0.803
Arjunone	−5.153	−2.701	−1.844	−1.973
beta-Sitosterol	−5.092	−2.409	1.034	−1.371
Casuarinin	−18.276	−9.216	−10.685	−9.678
Catechin	−5.665	−4.501	−4.202	−1.838
Ellagic acid	−6.023	−4.192	−2.769	-
Ethyl gallate	−6.776	−4.652	−3.616	−3.072
Gallic acid	−6.376	−4.95	−4.155	−1.816
Kaempferol	−5.498	−5.151	−3.026	−2.473
Leucocyanidin	−8.239	−3.649	−5.325	−3.955
Luteolin	−7.705	−5.137	−4.134	−2.103
Oleanolic acid	-	−0.464	−0.457	−0.539
Quercetin	−8.355	−5.794	−4.463	−1.79
Rutin	−10.668	−6.619	−7.787	−7.684
Terminic acid	-	-	−4.463	−1.066
Lisinopril	−7.722	−5.654	−3.573	−4.715
Phosphocholine	−5.401	−2.933	−4.565	−4.609
Losartan	−5.663	−4.363	−3.48	−3.632
Simvastatin	−5.286	−1.434	−1.732	−2.18
Losmapimod	−4.687	−3.133	−1.344	−1.582

**Table 2 molecules-28-01046-t002:** MM/GBSA profiles of casuarinin while interacting with four targeted proteins.

Target Proteins	∆G_Bind_ (kcal/mol)	∆G_vdW_ (kcal/mol)	∆G_Coulomb_ (kcal/mol)	∆G_H-bond_ (kcal/mol)	∆G_Lipo_ (kcal/mol)	∆G_Solv GB_ (kcal/mol)
4YAY	−79.43	−68.58	−89.27	−6.26	−24.07	103.86
4DLI	−14.72	−12.45	−44.26	−1.35	−4.83	48.91
1HW9	−34.20	−32.32	−44.05	−1.81	−7.76	50.75
1B09	−19.68	−14.01	−25.17	−1.63	−5.26	25.01

Coulomb—Coulomb energy. H-bond—hydrogen-bonding correction. Lipo—lipophilic energy, vdW—van der Waals energy.

**Table 3 molecules-28-01046-t003:** Drug-likeness and toxicity prediction of top-ranked phytocompound from *T. arjuna* and standard drugs.

Compounds	Drug-Likeness						Toxicity Prediction			
	cLogP (<5)	n_rot_ (<5)	MW (<500 Da)	HBD (<5)	HBA (<10)	Lipinski Rule	Hepato-Toxicity	Carcino- Genicity	Cyto- Toxicity	LD_50_ (mg/kg)
Casuarinin	−3.23	4	936.65	16	26	No	No	No	No	2170 (Class V)
Lisinopril	−1.46	13	405.49	4	7	Yes	No	No	No	8500 (Class VI)
Phosphocholine	−4.54	4	184.15	2	4	Yes	No	No	No	12,900 (Class VI)
Losartan	3.36	8	422.91	2	5	Yes	No	No	No	300 (Class III)
Simvastatin	3.77	7	418.57	1	5	Yes	No	No	No	1000 (Class IV)

clogP—measure of molecular hydrophobicity; n_rot_—number of rotatable bonds; MW—molecular weight; HBA—H-bond acceptor; HBD—H-bond donor; LD_50_—lethal dose.

**Table 4 molecules-28-01046-t004:** Details of target proteins and grid box coordinates for docking.

Target Proteins	Amino Acids	Resolution	Chain Selected for Docking	Grid Box Coordinates
Human Angiotensin receptor (PDB ID: 4YAY)	412	2.90 Å	Chain-A	x = −22.32; y = 6.81; z = 33.81
P38 Mitogen-activated protein kinase (MAPK, PDB ID: 4DLI)	360	1.91 Å	Chain-A	x = 24.22; y = −16.74; z = −10.1
HMG-Co A reductase (PDB ID: 1HW9)	467	2.33 Å	Chain-A	x = −5.94; y = −0.9; z = −19.46
Human C-reactive Protein (PDB ID: 1B09)	206	2.50 Å	Chain-A	x = −5.94; y = −0.9; z = −19.46

**Table 5 molecules-28-01046-t005:** Major phytocompounds present in various parts of *T. arjuna* for docking studies.

Sr. No	Compounds/Drugs	Formula	Compound ID
1	Arjunetin	C_36_H_58_O_10_	21152828
2	Arjunic acid	C_30_H_48_O_5_	15385516
3	Arjunolic acid	C_30_H_48_O_5_	73641
4	Arjunone	C_19_H_20_O_6_	14034821
5	Arjungenin	C_30_H_48_O_6_	12444386
6	β-sitosterol	C_29_H_50_O	222284
7	Casuarinin	C_41_H_28_O_26_	13834145
8	Ellagic acid	C_14_H_6_O_8_	5281855
9	Ethyl gallate	C_9_H_10_O_5_	13250
10	Gallic acid	C_7_H_6_O_5_	370
11	Luteolin	C_15_H_10_O_6_	5280445
12	Quercetin	C_15_H_10_O_7_	5280343
13	Terminic acid	C_30_H_48_O_4_	132568257
14	(+)-Catechin	C_15_H_14_O_6_	9064
15	Rutin	C_27_H_30_O_16_	5280805
16	Kaempferol	C_15_H_10_O_6_	5280863
17	Leucocyanidin	C_15_H_14_O_7_	71629
18	3-O-Methylellagic acid 3’-rhamnoside	C_21_H_18_O_12_	5319609
19	Oleanolic acid	C_30_H_48_O_3_	10494
20	Phosphocholine (PC)	C_5_H_15_NO_4_P^+^	1014
21	Simvastatin	C_25_H_38_O_5_	54454
24	Lisinopril	C_21_H_31_N_3_O_5_	5362119
25	Losartan	C_22_H_23_ClN_6_O	3961
26	Losmapimod	C_22_H_26_FN_3_O_2_	11552706

## Data Availability

Not applicable.

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
