# Peer review of "Multitarget Potential of Phytochemicals from Traditional Medicinal Tree, Terminalia arjuna (Roxb. ex DC.) Wight & Arnot as Potential Medicaments for Cardiovascular Disease: An In-Silico Approach"

_molecules, 2023, doi:10.3390/molecules28031046_

Round 1

Reviewer 1 Report

The manuscript entitled "Multitarget potential of phytochemicals from traditional medicinal tree, Terminalia arjuna (Roxb. ex DC.) Wight & Arnot as potential medicaments for cardiovascular disease: An in-silico approach" by Kumar at al. is a preliminary study and needs substantial improvement. There are technical mistakes in the study, therefore this study cannot be published in its current form.

Comments:

1.  Abstract and keywords are missing, and the whole write up needs improvement. English language should be checked by native speaker

2. The full form of CVDs should be written at first time (introduction, line 27)

3.  Targets are selected randomly, there is no proper involvement of the targets in appropriate pathways is discussed.

4.  A lay out or workflow scheme should be added in the methodology section

5. Line 37-38 “Any blockage of this system, including angiotensin-converting enzyme (ACE) inhibitors and angiotensin receptor antagonists, has become a first-line treatment for hypertensive target organ damage and progressive renal disease [4, 5]” should be rephrased

6. The role and pathway of 3-hydroxy-3-methylglutaryl-coenzyme A reductase (HMGCR), P38 kinases, and MAPK P38 in renin-angiotensin system (RAS) should be discussed in the introduction.

7. Several sentence in the introduction like “We have a working hypothesis that these compounds have the ability to bind with many cardiovascular target proteins;” line 82-83, and “Furthermore, by inserting a molecule non-covalently into the binding site of an object macromolecule, this technique increases systemic research by resulting in specific binding at each ligand's active sites [31-33].” Line 75-76 does not make any sense and wage.

8. In results, section 2.1. line 90-93 (casuarinin and rutin had a good binding affinity and better binding modes than that of selected standard drugs. Casuarinin showed binding energy of -9.678, -10.685, -9.216, and -18.276 kcal mol-1 with 1B09, 1HW9, 4DLI, and 4YAY proteins, respectively, while rutin showed binding energy of -7.684, -7.787, -6.619, and -10.668 kcal mol-1 with 1B09, 1HW9, 4DLI, and 4YAY proteins, respectively)

i)  Only 26 derivatives were selected for docking studies. Among those, 2 compounds showed good docking results. What was the purpose of selecting a small number of compounds?

ii) Two compounds showed promiscuous behavior by binding with all the target proteins, therefore the selectivity of the compounds are lost, which is highly unacceptable in medicinal chemistry.

9.  The use of AutoDock is purpose less here, this may be added just to show big data. When glide is used for docking, the proteins can directly be prepared on the glide. What was the need to use autodock tools for protein preparation?, and then docking on adt prepared proteins by glide? This is totally wage

10.  Ligands were minimized by Chem3D, but the minimization parameters are not discussed

11.  In the result section, a lot of significance of each method is discussed unnecessarily.

i) Like section 2.2 (Checking the RMSD of the protein can give knowledge into its auxiliary 3-D structural movement on a graph during the simulation. RMSD examination can demonstrate if the simulation has equilibrated-its changes towards the finish of the recreation are around some thermal energetically stable conformation)

ii) Accessible Surface Area (SASA) shows how much the Ligand is exposed to solvent when it interacts with the protein. If this value is less, then it is said that the ligand's interaction with protein is not affected by the solvent; if this value is high, then the solvent can pull out the ligand from the active site of the protein. Polar Surface Area (PSA) is the dissolvable available surface territory in a particle contributed uniquely by oxygen and nitrogen iotas.

iii) Section 2.3 (Binding energy calculation provides an insight into the ligand potential to strongly interact with the amino acids of the protein. The energy released (∆Gbind) due to bond formation, or rather interaction of the ligand with protein is in the form of binding energy and it determines the stability of any given protein-ligand complex. The free energy of a favorable reaction is negative.)

12. The docking was not validated by re-docking/cross docking of co-crystallized ligands.

13. The study is not supported by in vitro testing. At least two molecules (casuarinin and rutin) should be tested in vitro with all, or any one of the best targets. 

Author Response

Manuscript ID: Molecules-2126927

Title: Multitarget potential of phytochemicals from traditional medicinal tree, Terminalia arjuna (Roxb. ex DC.) Wight & Arnot as potential medicaments for cardiovascular disease: An in-silico approach

Comments:

  1. Abstract and keywords are missing, and the whole write up needs improvement. English language should be checked by native speaker.

Reply: Abstract and keywords have been added to the revised manuscript. English language was checked by native speaker.

  1. The full form of CVDs should be written at first time (introduction, line 27)

Reply: The full form of CVDs was added in the Introduction.

  1. Targets are selected randomly, there is no proper involvement of the targets in appropriate pathways is discussed.

Reply: Thanks for your comments. We have added the information regarding the target proteins in the revised manuscript.

  1. A lay out or workflow scheme should be added in the methodology section

Reply: We have added a workflow scheme in the methodology section in the revised manuscript.

  1. Line 37-38 “Any blockage of this system, including angiotensin-converting enzyme (ACE) inhibitors and angiotensin receptor antagonists, has become a first-line treatment for hypertensive target organ damage and progressive renal disease [4, 5]” should be rephrased.

Reply: We have rephrased the sentence in the revised manuscript.

  1. The role and pathway of 3-hydroxy-3-methylglutaryl-coenzyme A reductase (HMGCR), P38 kinases, and MAPK P38 in renin-angiotensin system (RAS) should be discussed in the introduction.

Reply: We have added the required information in the revised manuscript.

  1. Several sentence in the introduction like “We have a working hypothesis that these compounds have the ability to bind with many cardiovascular target proteins;” line 82-83, and “Furthermore, by inserting a molecule non-covalently into the binding site of an object macromolecule, this technique increases systemic research by resulting in specific binding at each ligand's active sites [31-33].” Line 75-76 does not make any sense and wage.

Reply: We have removed these lines from the revised manuscript.

  1. In results, section 2.1. line 90-93 (casuarinin and rutin had a good binding affinity and better binding modes than that of selected standard drugs. Casuarinin showed binding energy of -9.678, -10.685, -9.216, and -18.276 kcal mol-1 with 1B09, 1HW9, 4DLI, and 4YAY proteins, respectively, while rutin showed binding energy of -7.684, -7.787, -6.619, and -10.668 kcal mol-1 with 1B09, 1HW9, 4DLI, and 4YAY proteins, respectively)
  2. i)  Only 26 derivatives were selected for docking studies. Among those, 2 compounds showed good docking results. What was the purpose of selecting a small number of compounds?
  3. ii) Two compounds showed promiscuous behavior by binding with all the target proteins, therefore the selectivity of the compounds are lost, which is highly unacceptable in medicinal chemistry.

Reply: We have selected T. arjuna tree because of its cardioprotective property. Several studies have performed on its aqueous or ethanolic bark extract, but very limited studies showed which phytocompound of T. arjuna bark is responsible for its cardioprotective nature. Therefore, in the present study, only 26 phytocompounds of T. arjuna were selected against various protein targets. Out of these, casuarinin was found to show strong binding with all the selected protein targets. However, in vivo, in vitro and clinical studies are required to validate these findings.

  1. The use of AutoDock is purpose less here, this may be added just to show big data. When glide is used for docking, the proteins can directly be prepared on the glide. What was the need to use autodock tools for protein preparation? and then docking on adt prepared proteins by glide? This is totally wage.

Reply: We have checked the methodology and update the preparation of proteins as it was prepared in Glide.

  1. Ligands were minimized by Chem3D, but the minimization parameters are not discussed.

Reply: We have checked the methodology and update the preparation of ligands as it was prepared in Glide and the minimization parameters were also discussed.

  1. In the result section, a lot of significance of each method is discussed unnecessarily.
  2. i) Like section 2.2 (Checking the RMSD of the protein can give knowledge into its auxiliary 3-D structural movement on a graph during the simulation. RMSD examination can demonstrate if the simulation has equilibrated-its changes towards the finish of the recreation are around some thermal energetically stable conformation)
  3. ii) Accessible Surface Area (SASA) shows how much the Ligand is exposed to solvent when it interacts with the protein. If this value is less, then it is said that the ligand's interaction with protein is not affected by the solvent; if this value is high, then the solvent can pull out the ligand from the active site of the protein. Polar Surface Area (PSA) is the dissolvable available surface territory in a particle contributed uniquely by oxygen and nitrogen iotas.

iii) Section 2.3 (Binding energy calculation provides an insight into the ligand potential to strongly interact with the amino acids of the protein. The energy released (∆Gbind) due to bond formation, or rather interaction of the ligand with protein is in the form of binding energy and it determines the stability of any given protein-ligand complex. The free energy of a favorable reaction is negative.)

 Reply: We have removed the significance of methods in the revised manuscript.

  1. The docking was not validated by re-docking/cross docking of co-crystallized ligands.

Reply: We had not included the redocking data in the previous version to avoid disparity in the results, because the native ligand was absent in the 1B09 PDB entry.  We have added the redocking data to the supplementary section. The redocking data shows the appropriate redocking of native ligands with the respective targets. Furthermore, the docked phytochemicals showed better binding than the re-docked ligands.

  1. The study is not supported by in vitro testing. At least two molecules (casuarinin and rutin) should be tested in vitro with all, or any one of the best targets. 

Reply: The present study primarily based on in silico analysis of phytocompounds of Terminalia arjuna against various cardiovascular proteins. The results of the present study showed strong binding of casuarinin with all the selected target proteins. In future, we will also perform in vitro testing of casuarinin with selected protein targets

Reviewer 2 Report

the article with title " Multitarget potential of phytochemicals from traditional medicinal tree, Terminalia arjuna (Roxb. ex DC.) Wight & Arnot as potential medicaments for cardiovascular disease: An in-silico approach" show an interesting study as computational application on drugs as type of dry lab simulations but it need major revision in English and i did not find the abstract so please insert it      

Author Response

The article with title " Multitarget potential of phytochemicals from traditional medicinal tree, Terminalia arjuna (Roxb. ex DC.) Wight & Arnot as potential medicaments for cardiovascular disease: An in-silico approach" show an interesting study as computational application on drugs as type of dry lab simulations but it need major revision in English and i did not find the abstract so please insert it.

Reply: Thanks for your comments.  We have added the abstract in the revised manuscript and language of the manuscript have been revised.     

Round 2

Reviewer 1 Report

The manuscript entitled "Multitarget potential of phytochemicals from traditional medicinal tree, Terminalia arjuna (Roxb. ex DC.) Wight & Arnot as potential medicaments for cardiovascular disease: An in-silico approach" by Kumar et al has been revised according to the comments. Therefore this work is acceptable for publication after minor revision.

Comments:

The figures are unreadable and of lower resolution. For example:

In figure 1, the labels are overlapped.

In figure 2 and 3, resolution is quite low, and label font is very small.

Similarly figure 4 is completely unclear in its printable form.

The Y-axis labels of Figure 5 C and D is overlapped and unclear. 

Similarly, labels are overlapped and and cutted in figure s1 presented in supplementary information

Reviewer 2 Report

the article can be aceepted as it is